# Oilseed rape (*Brassica napus*) resistance to growth of *Leptosphaeria maculans* in leaves of young plants contributes to quantitative resistance in stems of adult plants

**Yong-Ju Huang**[1]*, **Sophie Paillard**[2], **Vinod Kumar**[2], **Graham J. King**[3], **Bruce D. L. Fitt**[1], **Régine Delourme**[2]

**1** School of Life and Medical Sciences, University of Hertfordshire, Hatfield, Hertfordshire, England, United Kingdom, **2** IGEPP, INRA, Agrocampus Ouest, Univ Rennes, BP, France, **3** Southern Cross University, Lismore, Australia

* y.huang8@herts.ac.uk

**Data Availability Statement:** All relevant data are within the manuscript and its Supporting Information files.

## Abstract

*Key message*: One QTL for resistance against *Leptosphaeria maculans* growth in leaves of young plants in controlled environments overlapped with one QTL detected in adult plants in field experiments.

The fungal pathogen *Leptosphaeria maculans* initially infects leaves of oilseed rape (*Brassica napus*) in autumn in Europe and then grows systemically from leaf lesions along the leaf petiole to the stem, where it causes damaging phoma stem canker (blackleg) in summer before harvest. Due to the difficulties of investigating resistance to *L. maculans* growth in leaves and petioles under field conditions, identification of quantitative resistance typically relies on end of season stem canker assessment on adult plants. To investigate whether quantitative resistance can be detected in young plants, we first selected nine representative DH (doubled haploid) lines from an oilseed rape DY ('Darmor-*bzh*' × 'Yudal') mapping population segregating for quantitative resistance against *L. maculans* for controlled environment experiment (CE). We observed a significant correlation between distance grown by *L. maculans* along the leaf petiole towards the stem ($r = 0.91$) in CE experiments and the severity of phoma stem canker in field experiments. To further investigate quantitative trait loci (QTL) related to resistance against growth of *L. maculans* in leaves of young plants in CE experiments, we selected 190 DH lines and compared the QTL detected in CE experiments with QTL related to stem canker severity in stems of adult plants in field experiments. Five QTL for resistance to *L. maculans* growth along the leaf petiole were detected; collectively they explained 35% of the variance. Two of these were also detected in leaf lesion area assessments and each explained 10–12% of the variance. One QTL on A02 co-localized with a QTL detected in stems of adult plants in field experiments. This suggests that resistance to the growth of *L. maculans* from leaves along the petioles towards the stems contributes to the quantitative resistance assessed in stems of adult plants in field experiments at the end of the growing season.

**Funding:** The UK Biotechnology and Biological Sciences Research Council (BBSRC, BB/I017585/1, M028348/1 and P00489X/1), the Innovate UK (102100 and 102641), AHDB Cereals & Oilseeds (RD-2140021105), the Department for Environment, Food and Rural Affairs (CH0104) (Defra), the Perry Foundation, the Felix Thornley Cobbold Agricultural Trust (5CXXXII) and the Chadacre Agricultural Trust (GRA 3/53) supported the work. The funders had no role in study design, data collection and analysis, decision to publish, or preparation of the manuscript.

**Competing interests:** The authors have declared that no competing interests exist.

## Introduction

Protecting crops from catastrophic yield losses caused by plant pathogens is a major goal of agriculture to safeguard global food security in response to growing concerns about food shortages and climate change [1–3]. Use of crop resistance is one of the most economical methods of controlling crop diseases, as it not only reduces input costs for farmers but also reduces risk of environmental damage. Crop resistance is particularly important for small-holder farmers who can not afford to buy expensive fungicides [3]. Furthermore, with a limited range of fungicides available and the development of fungicide insensitivity in pathogen populations [1, 4–7], identification of effective crop resistance is vital for sustainable crop production.

Plant resistance used to control crop diseases is generally classified as either major gene-determined qualitative resistance or minor gene-determined quantitative resistance [8–10]. Qualitative resistance, considered as 'complete' resistance, is usually controlled by single, dominant resistance (*R*) genes which are often effective in preventing pathogens from colonising plants [11–14]. However, *R* gene-mediated is race-specific; therefore, it is generally less durable than quantitative resistance because pathogens often evolve rapidly for virulence against the *R* genes [15–19]. By contrast, quantitative resistance is typically controlled by several genes with minor effects. Therefore, quantitative resistance is considered as incomplete or partial resistance that does not prevent pathogens from colonising plants but is able to decrease symptom severity and/or epidemic progress over time [10, 20–22]. Combining resistance QTL (quantitative trait loci) with complementary modes of action or combining quantitative resistance with major *R* genes have proved to be valuable strategies for breeding effective, potentially durable resistance [23]. Therefore, understanding quantitative resistance will contribute to deployment of crop cultivars with more durable resistance.

Phoma stem canker (blackleg) is an economically important disease of oilseed rape (*Brassica napus*, canola) worldwide, especially in Australia, North America and Europe [24, 25]. Epidemics of phoma stem canker are initiated by air-borne *Leptosphaeria maculans* ascospores released from pseudothecia produced on stem debris of previous crops [22, 26, 27]. In Europe, ascospores landing on leaf surfaces produce germ tubes that infect the leaves of winter oilseed rape in autumn (October/November), causing phoma leaf spot lesions [22, 28, 29]. From leaf lesions, *L. maculans* grows symptomlessly along the leaf midrib and petiole to reach the stem, causing damaging phoma stem cankers in spring/summer (April-July) that result in yield losses [12, 25, 30]. Both qualitative resistance and quantitative resistance have been identified in *B. napus* and related brassica species [9, 31, 32]. *R* gene-mediated resistance against *L. maculans* operates at the leaf infection stage through a defence response that causes cell death at the infection site, thus preventing *L. maculans* growth from leaf to stem and subsequently preventing stem canker development [14, 22, 33–35]. Quantitative resistance to *L. maculans* is a partial resistance that does not prevent leaf lesion development but decreases growth of *L. maculans* along leaf petioles and into stem tissues, therefore subsequently reduces the severity of stem cankers before harvest [22, 36, 37].

Compared to *R*-gene mediated resistance against *L. maculans*, quantitative resistance against *L. maculans* is less understood. One of the reasons is that *R* gene-mediated resistance against *L. maculans* can be selected effectively in young plants by observing necrosis symptoms after inoculation of cotyledons with well-characterised *L. maculans* isolates [14, 16, 33]. At least 15 *R* genes have been identified and two of them have been cloned [9, 14, 31, 38–40]. Due to the high evolutionary potential through sexual recombination, *L. maculans* can mutate to overcome *R*-gene mediated resistance within three cropping seasons of widespread use in commercial cultivars [15, 16, 18, 19]. Recent studies showed that combining *R*-gene resistance

with quantitative resistance in one crop cultivar provides a more robust strategy to control phoma stem canker [22, 30, 41].

Due to the long period of symptomless growth after initial leaf infection, it has been difficult to investigate quantitative resistance [21]. In Europe, it takes at least 6 months from the appearance of phoma leaf spots in autumn to the appearance of phoma stem canker in the following spring [26, 27]. It is difficult to investigate resistance to *L. maculans* growth in leaves and petioles or in stems before the appearance of stem canker symptoms under field conditions. Currently, selection of cultivars with quantitative resistance has relied on assessment of crop disease severity at the end of growing season [42–44]. Previous work showed that resistance against *L. maculans* growth in the leaf petiole towards the stem in young plants could be a component of quantitative resistance against *L. maculans* [36]. However, it is not clear whether the measurement of *L. maculans* growth in the leaf petioles of young plants in controlled environments can be used to detect QTL for resistance against *L. maculans* in adult plants in field conditions. Coincident resistance QTL detected in both controlled environment experiments in young plants and field experiments in adult plants would represent a valuable step towards identification of underlying resistance genes and provide a practical basis for marker-assisted resistance breeding. We were therefore keen to test the hypothesis that QTL for resistance against *L. maculans* can be detected in leaf petioles of young plants in controlled environments, and if so, to establish whether they co-localize with resistance QTL detected in stems of adult plants in field experiments. This paper reports work on detection of QTL for resistance against *L. maculans* growth in leaves of young plants in controlled environment experiments and compares them with QTL detected in stems of adult plants in field experiments using a doubled haploid (DH) segregating population.

## Materials and methods

### Relationship between *L. maculans* growth in leaves of young plants in controlled environment experiments and severity of phoma stem canker in stems of adult plants in field experiments (nine DH lines)

In field conditions in Europe, phoma stem cankers on adult plants in summer are usually initiated from phoma leaf lesions on young plants in autumn/winter [25, 27]. Therefore, to investigate whether the 'field' quantitative resistance against *L. maculans* assessed by scoring stem canker at the end of the growing season on adult plants in field experiments can be detected by assessing *L. maculans* growth in the leaves of young plants in controlled environments, nine DH lines with different levels of resistance to *L. maculans* were used. The DH lines were from an oilseed rape DY ('Darmor-*bzh*' × 'Yudal') mapping population; the parent 'Darmor-*bzh*' is winter oilseed rape with quantitative resistance against *L. maculans* and the resistance gene *Rlm9* while the parent 'Yudal' is a susceptible spring oilseed rape [42, 45]. Selection of the nine DH lines was based on stem canker severity assessed in 1995, 1996 and 2007 field experiments in France since there were good correlations between stem canker severities across these three years (S1 Fig; Kumar et al., 2018). The nine DH lines were classified from the most susceptible to the most resistant according to the disease severity index (DI): DY002, DY046, DY052, DY087, DY128, DY071, DY130, DY152 and DY283 [42, 44].

Plants were grown in pots (9 cm diameter) containing a peat-based compost mixed with a soluble fertiliser (WE Hewitt & Sons Ltd, UK). Plants were initially grown in a glasshouse (20–23˚C) and transferred to a controlled environment growth room (20ºC day/20˚C night, 12 h light/12 h darkness, light intensity 210 µmol $m^{-2}s^{-1}$) when they had three expanded leaves for inoculation. Ascospores of *L. maculans* from natural populations in the UK were used as inoculum. Although the parent 'Darmor-*bzh*' has resistance gene *Rlm9*, the *L. maculans*

populations throughout Europe have been virulent against *Rlm9* [11, 30, 46], so there was no *Rlm9* effect. The ascospore suspension of *L. maculans* was prepared from stem base debris of the UK susceptible oilseed cultivar Courage (naturally infected the stems collected after harvest in August 2007, stem pieces with mature pseudothecia were stored at -20˚C until required), using the method described by Huang et al. [29]. The concentration of ascospores was adjusted to $10^4$ ascospores mL$^{-1}$ using a haemocytometer slide.

To investigate the growth of *L. maculans* from leaf lesions along the leaf petioles towards the stems, leaf laminas were inoculated as described by Huang et al. (2014)[36]. Before inoculation, each leaf lamina was rubbed gently using a wet tissue and a 15 µL droplet of ascospore suspension was placed on the rubbed area. Each leaf had two inoculation sites, one on each side of the main vein as described by Huang et al. [36]. The first two fully expanded leaves of each plant were inoculated. After inoculation, plants in trays were covered with tray covers to maintain high relative humidity for 48h. Two experiments were done in a complete randomised design with 12 plants for each DH line.

The growth of *L. maculans* in the leaves was measured in terms of size of the phoma leaf spot lesions (i.e. lesion width, length and lesion area) and the growth along the leaf vein/petiole towards the stem (e.g. the visible distance grown along the petiole and the amount of *L. maculans* DNA in the petiole). At 18 days post inoculation (dpi), the inoculated leaves were detached at the place where the petiole joins the stem. The maximum length and width of each lesion were measured, then the lesion area was estimated by multiplying the lesion length by lesion width. The distance from the inoculation site on the leaf lamina to the furthest visible necrosis on the leaf petiole was measured on each leaf, then a piece of leaf petiole 8 cm long (measured from the inoculation site) was placed in a 15 mL tube to be freeze-dried for DNA extraction and quantitative PCR (qPCR). DNA was extracted from each individual leaf petiole using a DNA extraction kit (DNAMITE Plant Kit, Microzone Ltd, West Sussex, UK). The amount of *L. maculans* DNA in each leaf petiole was quantified using a SYBR green qPCR with the primers LmacF and LmacR [47] using a Stratagene Mx3000P quantitative PCR machine. The qPCR reaction and thermocycling profile were described by Huang et al. [36].

**Statistical analysis.** For leaf lamina inoculation experiments with the nine DH lines, the data for leaf lesion area, distance grown by *L. maculans* along the leaf petiole and the amount of *L. maculans* DNA in the leaf petiole were analysed using the restricted maximum likelihood procedure (REML). The analysis of variance components was done by fitting each of the response traits to a fixed model (constant + experiment + DH line + experiment × DH line) and a random model (experiment × block) to test the effects of experiment, DH line and the interaction between experiment and DH line. To investigate the relationship between *L. maculans* growth in leaves in controlled environments and severity of stem canker in field experiments, linear regressions were done for leaf lesion area, distance grown or amount of *L. maculans* DNA against the mean stem canker severity disease index on the nine DH lines assessed in 1995, 1996 and 2007 field experiments in France. All the analyses were done using GENSTAT statistical software 12$^{th}$ edition [48].

## Detection of QTL for resistance against *L. maculans* growth in leaves of young plants in controlled environment experiments (190 DH lines)

For the nine DH lines, significant correlations between by *L. maculans* growth in leaves of young plants in controlled environment experiments and the severity of phoma stem canker in field experiments were observed. To further investigate QTL related to resistance against growth of *L. maculans* in leaves of young plants, we selected 190 DH lines for leaf lamina inoculation. It was difficult to obtain large amounts of ascospore inoculum for the large

experiment, so conidia of *L. maculans* isolate ME24 were used since ME24 was virulent against *Rlm9* [36]. There were no differences between ascospores (used for the nine DH lines) and conidia of isolate ME24 in symptoms on Darmor or Yudal (i.e. both the ascospores and the conidia of ME24 caused typical phoma leaf lesions). Therefore, the conidia of ME24 were used as inoculum. Conidial suspensions of isolate ME24 were prepared from 12-day-old cultures on V8 agar and the concentration of conidia was adjusted to $10^7$ conidia mL$^{-1}$.

Plants of the 190 DH lines and their parents were grown in trays with 40 wells containing a peat-based compost mixed with a soluble fertiliser. Each tray was sown with 16 DH lines with two plants of each DH line. The middle row of wells was not sown to provide gaps between the DH lines. Plants were initially grown in a glasshouse (20–23˚C) and were transferred to a controlled environment growth room (20ºC day/20˚C night, 12 h light/12 h darkness, light intensity 210 μmol m$^{-2}$s$^{-1}$) for 24 h before inoculation. The experiments were designed in a resolvable alpha design with three replicates. The design allowed for sub-blocks of 16 lines, which were fitted into one tray. The allocation of trays to shelves (two trays per shelf) was done systematically following randomization. Plants were inoculated when they had three expanded leaves. Each leaf lamina was rubbed gently using a wet tissue and wounded using a sterile pin, then a 10 μL droplet of conidial suspension was placed over the wound. Each leaf had two inoculation sites, one on each side of the main vein. The first two fully expanded leaves of each plant were inoculated. After inoculation, plants were covered with tray covers to maintain high humidity for 72h since *L. maculans* conidia take longer to infect the plants than asco-spores [36].

At 21 dpi, the inoculated leaves were detached from the place where the petiole joins the stem. The maximum length and width of each lesion were measured, and the lesion area was estimated by multiplying the lesion length by lesion width. The distance from the inoculation site on the leaf lamina to the furthest visible necrosis on the leaf petiole was measured. To assess the effects of leaf lamina size and leaf petiole length on the growth of *L. maculans* in the leaf, the leaf lamina length and leaf petiole length of each leaf were also measured. The two experiments were done with the 190 DH lines.

**QTL detection in controlled environment experiments.** Data for each trait of the 190 DH lines and their parents were analysed using REML. Preliminary analysis of the data indicated that a square root transformation was required to stabilise the residual variance for leaf lesion area. Two REML analyses were done for each trait: (a) with DH line effects fitted as random effects using a fixed model and a random model, to enable calculation of heritability; (b) with DH line effects fitted as fixed effects using a fixed model and a random model, to provide predicted means for use in QTL analysis. The Pearson correlation coefficient was calculated using the predicted DH line means for all the traits.

A linkage map generated from the DY population comprising 3767 SNPs (one marker per genetic position) covering 2128.2 cM was used for QTL detection [49]. A multiple QTL mapping model was tested using the R/qtl package [50] for each variable as described by Kumar et al. [49]. For each variable, simple interval mapping (SIM) was done using the *scanone* function. The number of positions with a LOD score ≥ 2.5 was used to declare the number of cofactors in the composite interval analysis (CIM) done using the *cim* function. In the first model, positions with LOD scores ≥ 2.5 were included as QTL. An ANOVA was fitted to the multiple QTL model using the *fitqtl* function. QTL were retained in this first model when their effects were significant (α = 0.05). The *addqtl* function was then used to test for further significant QTLs, followed by an ANOVA fitted to the new model. To obtain the maximum likelihood estimates of the QTL positions, we then applied the *refineqtl* function, which used an iterative algorithm to refine the locations of QTL in our multiple QTL model. A final ANOVA was fitted to this multiple QTL model. Based on results of the ANOVA, the percentage of the

variation explained by the model and the $R^2$ of each QTL was assessed. The LOD value for each QTL was obtained using the *fiqtl* function. The confidence intervals of each QTL were assessed with a LOD decrease of one (*lodint* function). QTL positions and marker names at these positions, confidence intervals, percentage of variation explained by each QTL ($R^2$), allelic effect and favourable alleles were scored.

### Comparison of QTL detected in controlled environment experiments with those previously detected in field experiments

To investigate whether the resistance QTL detected in leaves of young plants in the controlled environment experiments co-localize with those QTL previously detected in stems of adult plants in field experiments, the positions of those QTL were compared. For QTL detected in field experiments, phoma stem canker severity data from 11 field experiments with three different mapping populations (S2 Table; [49]) were used. Field experiments with the DY segregating population were done in seven years, with five years (1995, 1996, 2007, 2011 and 2012) at Le Rheu, INRA, France [42, 44, 51] and two years (2008 and 2009) at Rothamsted Research, Harpenden, UK [52]. Field experiments were also done for the DS ('Darmor' x 'Samourai' DH) population in 1998 and 1999 [53] and for the DB ('Darmor' x 'Bristol' F$_{2:3}$) population in 2008 and 2010 [54] at Le Rheu, INRA, France. For QTL analysis, in addition to analysis for individual years for each mapping population, the combined data sets using BLUP (best linear unbiased predictions) estimations for these three mapping populations were also used (S3 Table). For QTL on the same chromosome (linkage group), their genetic and physical positions and support intervals were compared to identify QTL detected in both controlled environment experiments in young plants and field experiments in adult plants. Physical positions were deduced from the physical localization of the SNP markers on the *B. napus* cv. Darmor reference genome sequence assembly (version 4.1) [55, 56].

## Results

### Relationship between *L. maculans* growth in leaves of young plants in controlled environment experiments and severity of phoma stem canker in stems of adult plants in field experiments

For the nine DH lines, there were significant differences between DH lines for leaf lesion area ($P < 0.001$, 8 d.f.), distance grown along the leaf petiole ($P < 0.001$, 8 d.f.) and the amount of *L. maculans* DNA in the petiole ($P < 0.001$, 8 d.f.) (Table 1). There were no differences between experiments or interactions between experiment and DH line for leaf lesion area or distance grown. For the amount of *L. maculans* DNA in the petiole, there was no difference between experiments but there was an interaction between experiment and DH line; however, the Wald statistic was relatively small for the effect of interaction (33.1) compared to the main effect of DH line (526.8). The heritability was high for distance grown (0.88) and *L. maculans* DNA (0.96). The measurement of distance grown by *L. maculans* along the leaf petiole correlated well with the measurement of leaf lesion area ($r = 0.90$; $P < 0.001$) or the amount of *L. maculans* DNA in the petiole ($r = 0.89$; $P < 0.001$). There was a smaller correlation between the leaf lesion area and the amount of *L. maculans* DNA in the petiole ($r = 0.68$; $P < 0.01$).

There was a significant positive correlation between distance ($Gt$) grown by *L. maculans* along the leaf petiole of young plants in controlled environment experiments and mean stem canker disease index ($Di$) in stems of adult plants of the nine DH lines assessed in 1995, 1996 and 2007 field experiments in France ($Di = 2.69Gt—0.42$, $r = 0.91$) (Fig 1A). Similar

**Table 1. Measurements of *Leptosphaeria maculans* growth in leaves after inoculation of the leaf lamina in two controlled environment experiments and the mean phoma stem canker severity (DI-disease index) in 1995, 1996 and 2007 field experiments in France with nine doubled haploid (DH) lines from the *Brassica napus* DY ('Darmor-*bzh*' x 'Yudal') mapping population.**

| DH line | Measurement | | | DI [d] ± SE |
|---|---|---|---|---|
| | La [a] (cm²) ± SE | Gt [b] (cm) ± SE | DNA [c] (Log₁₀) ± SE | |
| DY002 | 3.28 ± 0.52 | 2.91 ± 0.14 | 1.00 ± 0.09 | 7.52 ± 0.32 |
| DY046 | 2.55 ± 0.33 | 2.55 ± 0.14 | 0.92 ± 0.05 | 6.27 ± 0.54 |
| DY052 | 2.65 ± 0.55 | 2.35 ± 0.19 | 0.95 ± 0.08 | 6.09 ± 0.33 |
| DY087 | 1.37 ± 0.24 | 1.87 ± 0.16 | 0.19 ± 0.12 | 5.59 ± 0.47 |
| DY128 | 2.55 ± 0.47 | 2.12 ± 0.13 | 0.29 ± 0.09 | 5.51 ± 0.40 |
| DY130 | 2.37 ± 0.24 | 2.05 ± 0.10 | 0.34 ± 0.11 | 4.49 ± 0.42 |
| DY071 | 2.20 ± 0.40 | 2.00 ± 0.18 | 0.81 ± 0.09 | 4.21 ± 0.79 |
| DY152 | 1.48 ± 0.45 | 1.60 ± 0.14 | -0.15 ± 0.08 | 4.15 ± 0.62 |
| DY283 | 1.38 ± 0.21 | 1.56 ± 0.12 | -0.44 ± 0.07 | 3.43 ± 0.65 |
| *P* value | <0.001 | <0.001 | <0.001 | <0.001 |
| SED | 0.51 | 0.18 | 0.13 | 0.44 |

[a] La- phoma leaf spot lesion area

[b] Gt- total distance grown by *L. maculans* along the leaf petiole from the inoculation site

[c] DNA—the amount of *L. maculans* DNA in leaf petiole (Log₁₀—transformed).

[d] DI—mean phoma stem canker severity (disease index) from field experiments in 1995, 1996 and 2007 in France. The stem canker was scored on a 0–9 scale, then the disease index was calculated using the following formula: Disease index (DI) = $[(N_0 \times 0) + (N_1 \times 1) + (N_2 \times 2) + \ldots + (N_9 \times 9)]/Nt$, where $N_0, N_1, N_2, \ldots N_9$ = the number of plants with a canker score of 0, 1, 2, …9, respectively, and Nt = the total number of plants assessed [42].

relationships were obtained for leaf lesion area (*La*) (*Di* = 1.48 *La* + 2, *r* = 0.76) (Fig 1B) or the amount of *L. maculans* DNA (*d*) in the leaf petiole (*Di* = 0.03*d* + 3.26, *r* = 0.77) (Fig 1C).

## Detection of QTL for resistance against *L. maculans* growth in leaves of young plants in controlled environment experiments

For the 190 DH lines, phoma leaf spot lesions developed on all inoculated leaves. For some of the DH lines, the inoculated leaves had turned yellow by 21 dpi (Fig 2). The frequency distributions for distance grown by *L. maculans* along the leaf petiole and the phoma leaf spot lesion area were continuous (Fig 3A and 3B). The continuous distributions suggested that the resistance against *L. maculans* growth in leaves of the DY population was quantitative and probably due to multiple loci with additive effects. There was a wide range in either the phoma leaf spot lesion area (0.19 to 1.31 cm²) or distance grown by *L. maculans* (1.05 to 3.17 cm) among the 190 DH lines. The broad sense heritability of lesion area (0.44) was greater than that of distance grown along the leaf petiole (0.39). There were significant positive correlations between different measurements of *L. maculans* growth in leaves of the 190 DH lines (Table 2). There was also a wide range in leaf petiole length (3.54–10.54 cm), leaf lamina length (4.48–9.13 cm) and total leaf length (8.06–18.64 cm) among the 190 DH lines. The distributions of leaf petiole length (Fig 3C) or leaf lamina length (data not presented) were also continuous. The heritability of leaf petiole length (0.81) and leaf lamina length (0.74) was high. These traits were highly correlated with each other and moderately correlated with different measurements of *L. maculans* growth in leaves (Table 2).

Five QTL involved in resistance to *L. maculans* growth along the leaf petiole (CE-Gt) were detected on linkage groups A02, A03, A10, C01 and C09, respectively (Table 3). The estimated phenotypic variation explained by individual QTL ranged from 3 to 10% and they collectively explained 35% of the variance. In addition, one weak QTL was detected on A04. Two QTL

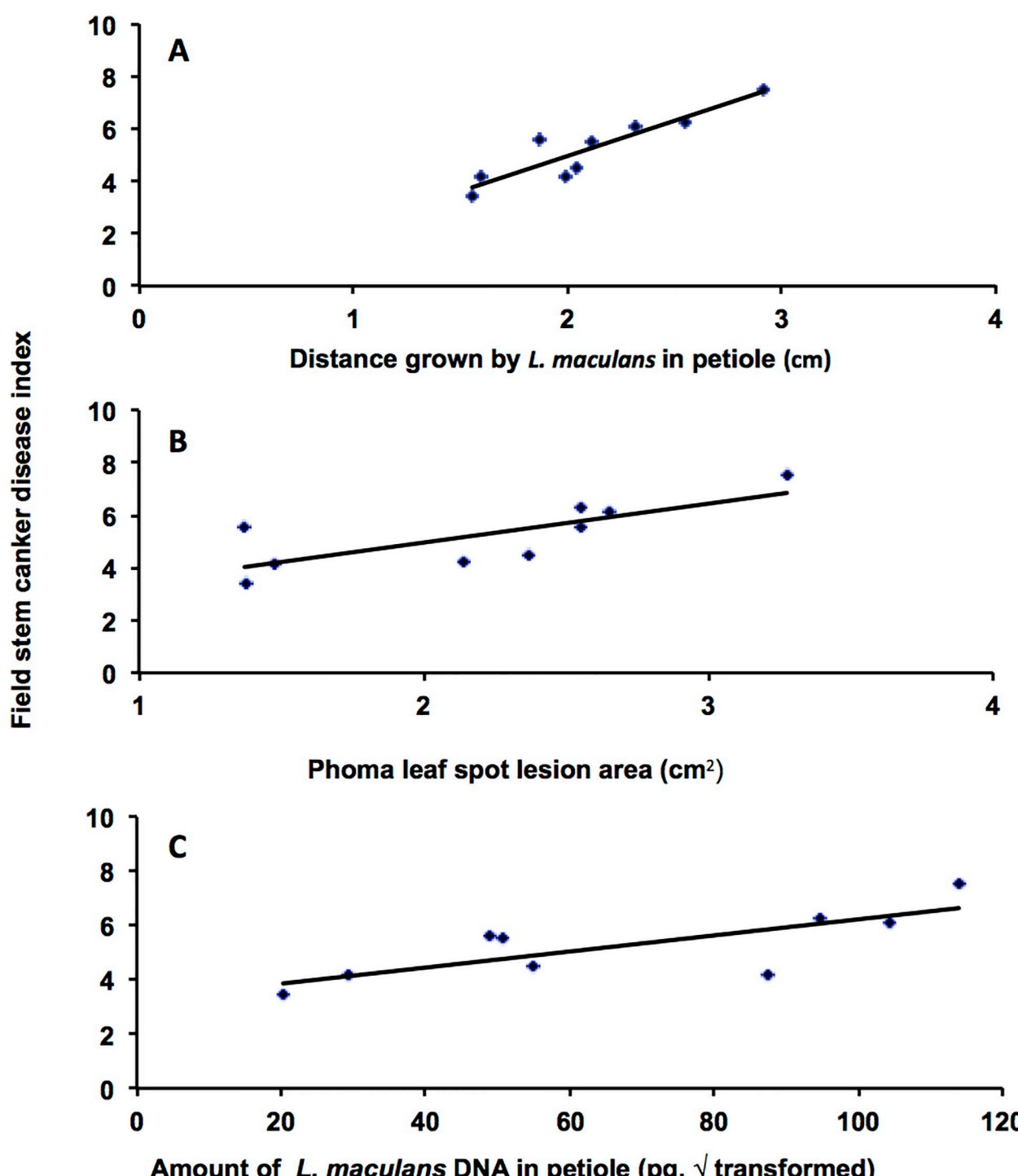

**Fig 1. Relationships between *Leptosphaeria maculans* growth in young plants and phoma stem canker severity in adult plants.** The relationships between distance grown by *L. maculans* in the leaf petiole (*Dt*) ($Di = 2.69Dt -0.42$, $r = 0.91$) (A), phoma leaf lesion area (*La*) ($Di = 1.48La +2$, $r = 0.76$) (B) or amount of *L. maculans* DNA in petiole (*d*) ($Di = 0.03d +3.26$, $r = 0.77$) (C) of young plants in controlled environment experiments and the mean phoma stem canker severity (*Di*; disease index) in winter oilseed rape field experiments in 1995, 1996 and 2007 with nine doubled haploid (DH) lines from the *Brassica napus* DY ('Darmor-*bzh*' x 'Yudal') mapping population assessed by linear regression. Data for controlled environment experiments are means of data from two replicate experiments.

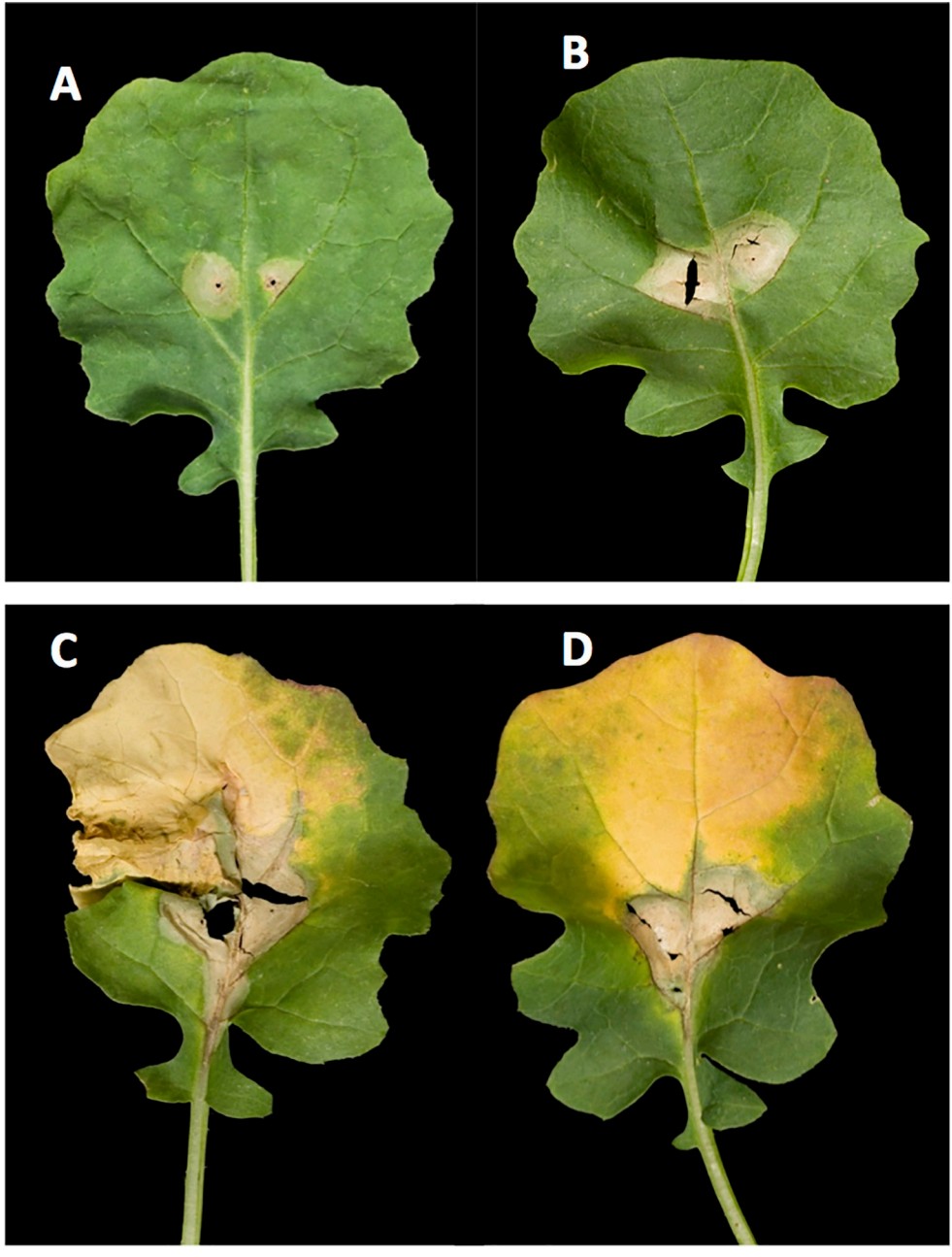

**Fig 2. Phoma leaf spot symptoms produced on different DH lines in controlled environment experiments.** Phoma leaf spot symptoms on leaves of four doubled haploid (DH) lines DY413 (A), DY087 (B), DY048 (C) and DY001 (D) inoculated with conidia of *Leptosphaeria maculans* isolate ME24 at 21 days post inoculation.

that collectively explained 22% of the variance in phoma leaf spot lesion area (CE-La) were detected on A02 and C01, and were located within support intervals for the two QTL related to *L. maculans* distance grown along the leaf petiole (Fig 4). The individual QTL each explained 10–12% of the variance (Table 3). The resistance alleles were contributed by 'Darmor-*bzh*' for QTL on A02, A03 and A10 and by 'Yudal' for QTL on A04, C01 and C09.

To investigate whether there were any correlations between leaf lamina size or leaf petiole length and the growth of *L. maculans* in the leaf, the QTL related to leaf lamina length and leaf

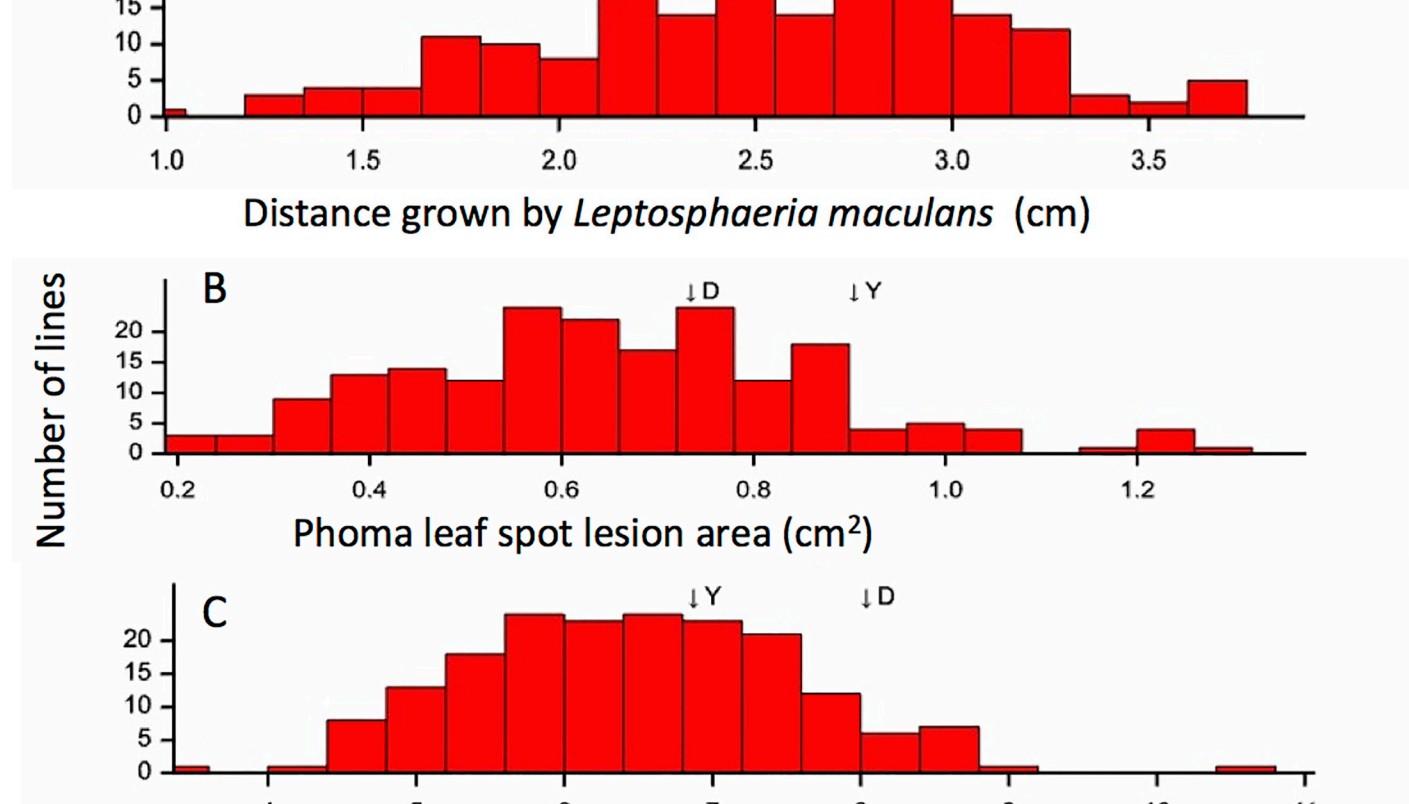

**Fig 3. Frequency distributions.** Frequency (number of DH lines) distributions of different variables measured for the 190 lines from the *Brassica napus* DY ('Darmor-*bzh*' x 'Yudal') mapping population in two controlled environment experiments. A, distance grown by *Leptosphaeria maculans* along the leaf petiole from the inoculation site (cm); B, phoma leaf spot lesion area (cm²); C, leaf petiole length (cm). Arrows indicate the mapping population parents (D = Darmor-*bzh*, Y = Yudal). Data are means of two controlled environment experiments.

petiole length were also analysed. Six QTL related to leaf lamina length (CE-Lal) were detected and distributed over six linkage groups (Table 4). The estimated phenotypic variation explained by individual QTL ranged from 5 to 31% and all QTL collectively explained 59% of the variation. A strong effect QTL was detected on A06 where the dwarf gene *bzh* was located. The alleles for smaller leaf lamina length were from parent 'Darmor-*bzh*' on A06 and A10 and from parent 'Yudal' for the other four QTL (Table 4). Six QTL related to leaf petiole length (CE-Pl) were detected and distributed over six linkage groups (Table 4). The estimated phenotypic variation explained by individual QTL ranged from 4 to 40% and all QTL collectively explained 57% of the variation. A strong effect QTL related to leaf petiole length was detected on A06 where the dwarf gene *bzh* was located. The alleles for shorter petiole length were from parent 'Darmor-*bzh*' on A06 and A09 and from parent 'Yudal' for the other four QTL (Table 4). QTL on A01, A06 and C06 related to leaf lamina length and leaf petiole length were at the same location and support intervals overlapped for the other QTL. When compared with QTL detected for leaf lesion length or distance grown by *L. maculans* in the leaf petiole

**Table 2. Correlation coefficient values for relationships between different measurements of *Leptosphaeria maculans* growth in leaves and for leaf size of 190 doubled haploid (DH) lines from the *Brassica napus* DY ('Darmor-*bzh*' x 'Yudal') mapping population in two controlled environment experiments.**

| Measurement | Correlation coefficient | | | | | | |
| --- | --- | --- | --- | --- | --- | --- | --- |
| | Distance grown | Lesion length | Lesion width | Lesion area | Lamina length | Petiole length | Total leaf length |
| Distance grown | 1.00 | | | | | | |
| Lesion length | 0.82*** | 1.00 | | | | | |
| Lesion width | 0.70*** | 0.85*** | 1.00 | | | | |
| Lesion area | 0.78*** | 0.96*** | 0.95*** | 1.00 | | | |
| Lamina length | 0.17* | 0.17* | 0.21** | 0.21** | 1.00 | | |
| Petiole length | 0.12<sup>NS</sup> | 0.11 <sup>NS</sup> | 0.16* | 0.16* | 0.80*** | 1.00 | |
| Total leaf length | 0.15* | 0.14 <sup>NS</sup> | 0.19* | 0.19* | 0.93*** | 0.96*** | 1.00 |

\* $P< 0.05$

\*\* $P< 0.01$

\*\*\* $P< 0.001$

[NS] Not significant

with QTL detected for leaf laminar length or petiole length, only a partial overlap with a QTL on A10 for leaf lamina length was found. No other co-localization was detected.

## Comparison of QTL detected in controlled environment experiments with those previously detected in field experiments

All the QTL detected based on the 190 DH lines from the DY ('Darmor-*bzh*' x 'Yudal' DH) population in the controlled environment experiments were compared with QTL detected in field experiments with the DY (seven years), DB ('Darmor' x 'Bristol' $F_{2:3}$) (two years) and DS ('Darmor' x 'Samourai' DH) (two years) populations. The comparisons represented either individual years or the combined data sets using BLUP estimations (S4 Table). The QTL on A02 detected for distance grown along the leaf petiole and leaf lesion area co-localized with QTL detected in field experiments in 2007 and 2012, and was also close to the QTL detected using BLUP for the DY population (Table 3; Fig 4). Furthermore, this QTL was also detected in 1998 and BLUP for the DS population (S4 Table; Fig 4). In all cases, the resistance allele was contributed by 'Darmor-*bzh*'. A QTL detected on A04 for distance grown along the leaf petiole co-localized with a QTL detected in a field experiment in 2008 and 2011 for the DY population and with BLUP QTL found in the DB population (S4 Table) but the resistance allele was not contributed by the same parent (Fig 4; Table 3). All the other QTL from controlled environment experiments did not colocalize with QTL detected in field experiments either in individual years or in BLUP. A QTL on A03 was detected in 2007, 2012 and BLUP with the DY population and in 2008 with the DB population (S4 Table), but the positions were different from the QTL detected in controlled environment experiments (Table 3; S2 Fig). Similarly, the QTL on C01 detected in field experiments was in a different position from that detected in controlled environment experiments (Table 3; Fig 4). However, the resistant allele of the QTL on C01 was from 'Yudal' for both controlled environment experiments and field experiments.

## Discussion

Results of this study show that it is possible to detect QTL for resistance to *L. maculans* in young plants by investigating the growth of the pathogen from leaf lesions along the leaf petiole towards the stem or by measuring leaf lesion area in controlled environment experiments. Five QTL for resistance to *L. maculans* growth along the leaf petiole were detected and two of

**Table 3. QTL for resistance against *Leptosphaeria maculans* detected in controlled environment experiments and neighbouring QTL detected in winter oilseed rape field experiments with the *Brassica napus* DY ('Darmor-*bzh*' x 'Yudal') mapping population.**

| Expt-trait [a] | LG [b] | Locus [c] | Position (cM) [c] | Support interval (cM) | Physical support interval (bp) | Effect [d] | LOD [e] | $R^2$ (%) [f] |
|---|---|---|---|---|---|---|---|---|
| CE-Gt | A02 | Bn-A02-p3074016 | 9.3 | 4.9–41.1 | 225 678–3 636 730 | 0.05 | 3.37 | 5.6 |
| BLUP-DI | A02 | cA02.loc1 | 1.0 | 0–3.8 | 12 296–87 358 | 0.16 | 13.84 | 6.0 |
| 2007-DI | A02 | BS012929 | 14.6 | 7.7–16.7 | 262 823–965 813 | 0.36 | 11.50 | 8.2 |
| 2012-DI | A02 | Bn-A02-p4243717 | 24.2 | 16.1–27.9 | 869 834–2 001 616 | 0.54 | 9.39 | 7.3 |
| CE-La | A02 | scaffoldv4_207_269733 | 35.4 | 7.7–40.5 | 337 265–3 775 620 | 0.02 | 5.1 | 10.3 |
| CE-Gt | A03 | Bn-A03-p6765024 | 51 | 47.6–54.9 | 1 614 358–9 727 949 | 0.05 | 3.0 | 5.0 |
| 2012-DI | A03 | scaffoldv4_46_962259 | 69.1 | 67.9–71.8 | 10 407 494–11 386 559 | -0.24 | 4.31 | 3.2 |
| CE-Gt | A04 | Bn-A04-p13122589 | 33.2 | 28.4–41.5 | 13 192 678–15 006 424 | -0.45 | 1.73 | 2.8 |
| 2011-DI | A04 | Bn-A04-p11794986 | 25.2 | 22.5–27.3 | 12 640 632–13 146 945 | 0.31 | 11.6 | 11.5 |
| 2008-DI | A04 | cA04.loc32 | 32 | 19.3–41.5 | 11 885 058–14 976 156 | 0.31 | 3.18 | 8.8 |
| BLUP-DI | A04 | Bn-A04-p11244643 | 22.4 | 21.2–25.9 | 12 211 710–12 876 658 | 0.34 | 22.77 | 10.7 |
| CE-Gt | A10 | Bn-C9-p51689508 | 56.7 | 33.6–64.5 | 13 391 605–15 935 120 | 0.06 | 3.8 | 6.3 |
| BLUP-DI | C01 | cC01.loc15 | 15 | 3–20.1 | 806 911–2 469 521 | -0.22 | 5.18 | 2.09 |
| 2008-DI | C01 | Bn-C1-p1914447 | 10.6 | 0–18.2 | 82 561–2 263 603 | -0.28 | 3.15 | 8.7 |
| CE-Gt | C01 | Bn-*scaffold02163-p1951 | 31.1 | 23.8–52.6 | 3 092 187–11 992 257 | -0.07 | 5.98 | 10.2 |
| CE-La | C01 | Bn-A01-p7150748 | 45.6 | 37.5–53.8 | 7 371 451–12 494 273 | -0.02 | 5.9 | 12.1 |
| CE-Gt | C09 | scaffoldv4_130_1134724 | 15.7 | 0.0–23.3 | 0–2 357 048 | -0.05 | 3.16 | 5.2 |

[a]Expt-trait, for experiments, CE- controlled environment experiments, the years 1995 to 2012 are the years when the field experiments were assessed for phoma stem canker severity, BLUP (best linear unbiased predictions)—estimation of combined data for seven years; for traits, Gt–total distance grown by *L. maculans* along the leaf petiole from the inoculation site, DI–disease severity index, La- phoma leaf spot lesion area.

[b]LG, the linkage groups, are named according to *Brassica napus* A01–A10 and C01–C09 designations by the Multinational *Brassica* Genome Project Steering Committee.

(http://www.brassica.info/information/lg_assigments.htm)

[c]The marker closest to the position of maximum effect of the QTL.

[d]The additive effect.

[e]Test statistic value for QTL, logarithm of the odds.

[f]Proportion (%) of the phenotypic variation explained by the QTL.

these were also detected by measurement of leaf lesion area. This is the first reported study investigating QTL for resistance against *L. maculans* growth in leaves of young plants in controlled environments. In Europe under natural conditions, the fungal pathogen *L. maculans* initially infects leaves of oilseed rape in autumn and then grows systemically from leaf lesions along the leaf petiole to the stem, causing damaging phoma stem canker (blackleg) in the following summer [9, 27, 57]. Traditionally, selection for quantitative resistance to the phoma stem canker pathogen has relied on winter oilseed rape field experiments in which phoma stem canker severity is assessed at the end of the cropping season before harvest [22, 42, 44]. Therefore, quantitative resistance against *L. maculans* has also been referred to as 'adult plant resistance' [9]. Because *L. maculans* has a long period of symptomless growth between the initial development of phoma spot lesions on leaves and the appearance of canker symptoms on stems, it has been difficult to investigate quantitative resistance to *L. maculans* growth in leaf petioles and stems under field conditions. Results of this work provide evidence that components of quantitative resistance to *L. maculans* can be detected in controlled conditions.

In the comparison of QTL for resistance against *L. maculans* growth in leaves of young plants in controlled environments and QTL for resistance against *L. maculans* growth in stems of adult plants in field experiments, one QTL on A02 was found to co-localize with a QTL

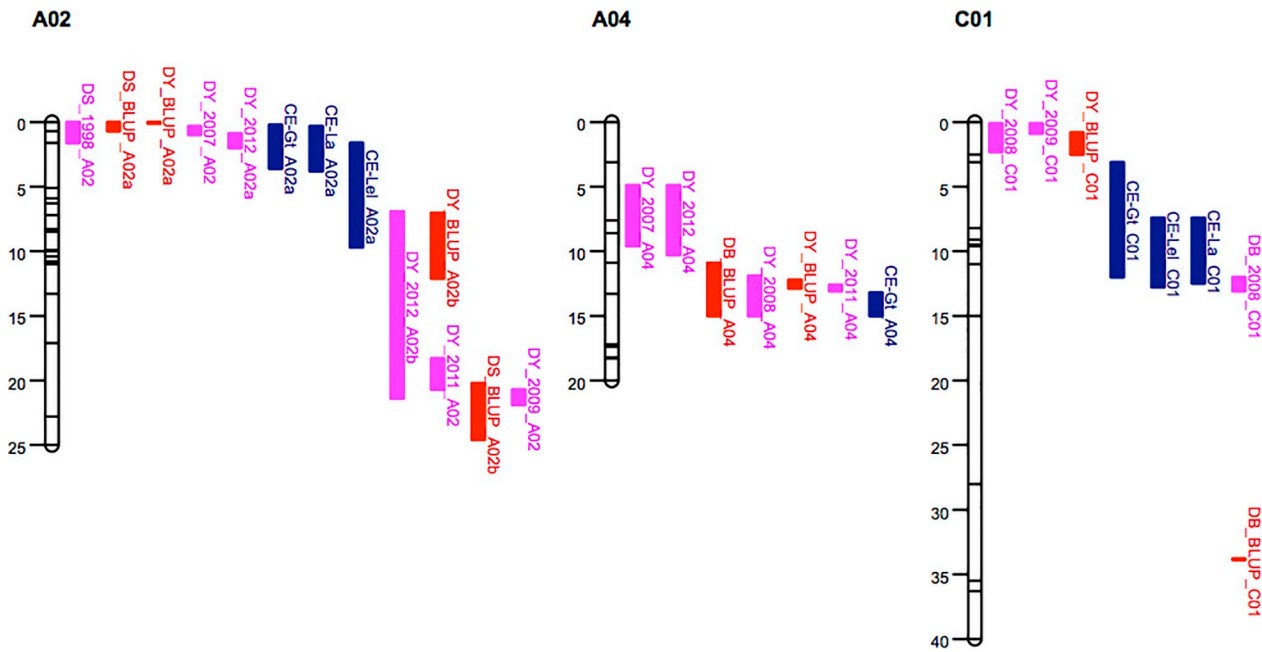

**Fig 4. Comparison of the physical position of QTL.** Comparison of the physical position (in Mb) of QTL on A02, A04 and C01 for resistance against *Leptosphaeria maculans* detected in controlled environment experiments (CE) in young plants with QTL detected in field experiments in adult plants. CE experiments were done with 190 lines from the *Brassica napus* DY ('Darmor-*bzh*' x 'Yudal' DH) population. Field experiments were done with DY population in 1995, 1996, 2007, 2008, 2009, 2011 and 2012, with DB ('Darmor' x 'Bristol' F$_{2:3}$) population in 2008 and 2010 and with DS ('Darmor' x 'Samourai' DH) population in 1998 and 1999 (for details see S1 and S2 Tables). QTL from CE experiments are in blue; QTL from individual years in field experiments are in pink; QQTL from combined years of field experiments using BLUP (best linear unbiased predictions) estimations are in orange.

detected in field experiments in 2007 and 2012 and was very close to the QTL detected with BLUP. Furthermore, this QTL was also detected in the same population after field phenotyping in Australia [58] and in another mapping population ('Darmor' x 'Samourai' DH) that shares the same source of quantitative resistance. Consistent detection of this QTL on A02 by measuring *L. maculans* growth along the leaf petiole or the size of the leaf lesion in controlled environment experiments or by measuring the stem canker severity in field experiments suggests that this QTL might operate to control the rate of *L. maculans* growth from the leaf to the stem, leading to reduced stem canker severity in adult plants before harvest. In other words, resistance to the growth of *L. maculans* in leaves of young plants contributes to quantitative resistance in stems of adult plants. Furthermore, this QTL was not affected by petiole length since no QTL for petiole length was detected in this region. This suggests that the growth rate of *L. maculans* along the leaf petiole towards the stem might be more important than the petiole length. This is partly supported by the poor correlations between *L. maculans* growth in the leaf petiole and the leaf laminar length or the leaf petiole length (Table 2). Interestingly, although a QTL on C01 detected in controlled environment experiments was at a different position from that detected in field experiments [49], the resistance allele of the QTL was from the susceptible parent 'Yudal' for both controlled environment experiments and field experiments. This suggests that the contribution from 'Yudal' to this QTL might operate to slow down the growth of *L. maculans* from the leaf to the stem and subsequently reduce stem canker severity in adult plants before harvest. More precise investigations are needed to test whether the QTL are the same, which would provide further evidence that resistance to the growth of *L. maculans* in leaves of young plants contributes to quantitative resistance in stems of adult plants.

**Table 4. Information about QTL for leaf lamina size and leaf petiole length detected by composite interval mapping in two controlled environment experiments (CE) with 190 doubled haploid (DH) lines from the *Brassica napus* DY ('Darmor-*bzh*' x 'Yudal') mapping population.**

| Trait | LG[a] | Locus[b] | Position (cM)[b] | Support interval (cM) | Physical support interval (bp) | LOD[c] | Effect[d] | $R^2$ (%)[e] |
|---|---|---|---|---|---|---|---|---|
| CE-Lal[f] | A01 | Bn-A01-p4246586 | 28.7 | 10.1–33.7 | 1497430–4638630 | 4 | -0.12 | 4.5 |
| CE-Lal | A03 | BS006552 | 3.6 | 0–19.1 | 13519–2234749 | 4.3 | -0.13 | 4.8 |
| CE-Lal | A06 | scaffoldv4_290_297119 | 113.5 | 112.9–115.2 | 22839484–23104279 | 21.9 | 0.34 | 30.7 |
| CE-Lal | A10 | Bn-A10-p10748653 | 23.5 | 19.6–49.7 | 11776434–14967637 | 5.3 | 0.15 | 6.0 |
| CE-Lal | C03 | Bn-C3-p29046921 | 111.9 | 103–116.4 | 23140183–30313620 | 4 | -0.12 | 4.5 |
| CE-Lal | C06 | cC06.loc62 | 62 | 60.5–66.7 | 25409207–31406603 | 7.6 | -0.16 | 8.9 |
| CE-Pl[g] | A01 | scaffoldv4_9817_2019 | 21.8 | 15.4–29 | 1850243–3984116 | 3.8 | -0.17 | 3.9 |
| CE-Pl | A03 | scaffoldv4_29_1342774 | 19.1 | 0–27.7 | 13519–2979192 | 2.8 | -0.15 | 2.8 |
| CE-Pl | A06 | scaffoldv4_290_297119 | 113.5 | 112.9–115.2 | 22839484–23104279 | 28.3 | 0.57 | 39.7 |
| CE-Pl | A09 | scaffoldv4_225_749028 | 103.3 | 99.4–127.2 | 28142060–33861569 | 3.8 | 0.26 | 3.9 |
| CE-Pl | C03 | Bn-C3-p54103303 | 147.5 | 0–148.7 | 59682–50147047 | 3 | -0.24 | 3.1 |
| CE-Pl | C06 | Bn-C6-p13174110 | 57.5 | 53.8–62.6 | 21940179–26892582 | 3.3 | -0.12 | 3.4 |

[a]LG, the linkage groups, are named according to *Brassica napus* A01–A10 and C01–C09 designations by the Multinational *Brassica* Genome Project Steering Committee (http://www.brassica.info/information/lg_assigments.htm)

[b]The marker closest to the position of maximum effect of the QTL

[c]Test statistic value for QTL

[d]The additive effect

[e]Proportion (%) of the phenotypic variation explained by the QTL

[f]Lal- the leaf lamina length

[g]Pl- the leaf petiole length

Although a QTL on A04 detected in controlled environment experiments for distance grown along leaf petiole co-localized with a QTL detected in the field experiment in 2008 and with BLUP QTL for the DB ('Darmor' x 'Bristol' $F_{2:3}$) population [49], the resistance allele was contributed by 'Yudal' (e.g. resistance to the growth of *L. maculans* in leaves) for controlled environment experiments and by 'Darmor-*bzh*' (e.g. resistance to the growth of *L. maculans* in stems) for field experiments. These results also suggest that the QTL operating in the leaf petiole (by slowing down the growth of *L. maculans* towards the stem) may be different from the QTL operating in the stem (by slowing down the growth of *L. maculans* in stems to reduce stem canker severity). This is supported by the evidence that QTL on A03, A10, C01 and C09 detected in leaves of young plants in controlled environment experiments are at different positions from those detected in stems of adult plans in field experiments (Fig 4; S1 Fig). Even though QTL operating in leaf petiole can delay the time when *L. maculans* reaches the stem, once *L. maculans* has reached the stem, the weather conditions and the QTL operating in the stem are vital in reducing the severity of stem canker. Recent work showed that the increased temperature during stem canker development stage (April–June; i.e. after *L. maculans* has reached the stem) is associated with increased phoma stem canker severity before harvest [30]. Furthermore, due to the interactions with environment, QTL for quantitative resistance are often inconsistent over locations or seasons [10, 59]. Our previous work showed that many QTL were sensitive to environmental factors [49, 52, 60]. Therefore, quantitative resistance in stems of adult plants in field experiments (assessed at the end of growing season) results not only from resistance to growth of *L. maculans* in the leaves but also from resistance to growth of *L. maculans* in the stems. Work on spring oilseed rape also showed that quantitative resistance against growth of *L. maculans* in stems of adult plants was partly contributed by the growth of *L. maculans* in cotyledons of seedlings [35, 61]. Only one QTL on C06 related to

resistance against growth of *L. maculans* in cotyledons detected in controlled environment experiments was found to co-localize with a QTL detected in stems of adult plants in field experiments [61] and might correspond to QTL reported in Kumar et al. [49]. Although cotyledon infection is observed in spring oilseed rape in Australia [26, 62], it is not common in Europe where the main source of phoma stem canker is from leaf lesions developed in autumn [25, 26]. Therefore, detection of QTL related to *L. maculans* growth in leaf petioles is close to what is happening in the field conditions in Europe.

Previous work showed that quantitative resistance to *L. maculans* could be assessed at two stages: stage 1- growth of the pathogen along leaf petioles towards the stem; and stage 2-growth in stem tissues to produce stem canker symptoms [36]. We showed in this work that resistance QTL related to *L. maculans* growth at stage 1 can be detected in controlled environments. Assessing resistance QTL in leaves of young plants (*L. maculans* growth stage 1) in controlled environment experiments is rapid; it takes 40–45 days from sowing to finishing the assessment for leaf lamina inoculation experiments. By contrast, assessing resistance QTL in stems of adult plants in field experiment before harvest takes 10–11 months (from sowing to harvest) for winter oilseed rape. Another advantage is that detection of QTL in controlled conditions can avoid the influence of fluctuations in natural weather conditions. However, only a small number of QTL detected in controlled environment experiments co-localized with QTL detected in field experiments. This suggests that only some components of quantitative resistance as assessed in the field might be detected in controlled environment experiments. The duration of *L. maculans* growth from the leaf petiole to the stem and *L. maculans* growth in the stem are other potential components of quantitative resistance. There is a need to investigate resistance QTL related to *L. maculans* growth in stem tissues to produce stem canker symptoms (*L. maculans* growth stage 2) and compare the QTL detected at both growth stages in controlled conditions with QTL detected in field conditions, which will help to understand components of quantitative resistance assessed at the end of the growing season in field experiments.

Only one isolate of *L. maculans* was used in the controlled environment experiments with the 190 DH lines for QTL detection, while in field conditions the plants were exposed to all the available pathogen races. This may be one of the reasons why only one QTL was co-localized with a QTL detected in stems of adult plants in field conditions. Although quantitative resistance against *L. maculans* is considered race non-specific, there may be differences between isolates in their aggressiveness and thus their ability to cause severe stem canker. There is a need to test more isolates or use a mixture of different isolates to investigate whether more common QTL can be detected in both controlled environments and in field experiments, which will help to develop markers for breeding cultivars with durable quantitative resistance.

## Supporting information

**S1 Table. Disease severity index and QTL detected in the nine DH (doubled haploid) lines from the Brassica napus DY ('Darmor-bzh' x 'Yudal') mapping population in seven field experiments.**
(XLSX)

**S2 Table. List of eleven field experiments which produced QTL based on the phoma stem canker severity data (Kumar et al. 2018) used for comparison with QTL detected in controlled environment experiments with those for 190 DH lines from the DY population.**
(DOCX)

**S3 Table. Phenotypic and genotypic data for the Brassica napus DY ('Darmor-bzh' x 'Yudal') mapping population used for detection of QTL for resistance against Leptosphaeria maculans in field experiments and controlled environment experiments.**
(XLSX)

**S4 Table. Linkage groups (LG) where QTL for resistance against Leptosphaeria maculans detected in controlled environment experiments with QTL detected in winter oilseed rape field experiments with the Brassica napus DY ('Darmor-bzh' x 'Yudal'), DB ('Darmor' x 'Bristol' F2:3) or DS ('Darmor' x 'Samourai' DH) population.**
(DOCX)

**S1 Fig. Correlations between disease severity index in different field experiments.** Correlations between phoma stem canker disease index in winter oilseed rape field experiments in 1995, 1996, 2007, 2008, 2009, 2011 and 2012 with doubled haploid (DH) lines from the *Brassica napus* DY ('Darmor-*bzh*' x 'Yudal') mapping population. For details of these field experiments see S2 Table.
(DOCX)

**S2 Fig. Comparison of the physical position of QTL.** Comparison of the physical position (in Mb) of QTL on A03, A10 and C09 for resistance against *Leptosphaeria maculans* detected in controlled environment experiments (CE) in young plants with QTL detected in field experiments in adult plants. CE experiments were done with 190 lines from the *Brassica napus* DY ('Darmor-*bzh*' x 'Yudal' DH) population. Field experiments were done with the DY population in 1995, 1996, 2007, 2008, 2009, 2011 and 2012, with the DB ('Darmor' x 'Bristol' $F_{2:3}$) population in 2008 and 2010 and with the DS ('Darmor' x 'Samourai' DH) population in 1998 and 1999 (for details see S1 & S2 Tables). QTL from CE experiments are in blue; QTL from individual year in field experiments are in pink; QTL from combined years of field experiments using BLUP (best linear unbiased predictions) estimation are in orange.
(DOCX)

## Acknowledgments

We thank Aiming Qi and Sue Welham for advice on experimental design and help with statistical data analysis.

## Author Contributions

**Conceptualization:** Yong-Ju Huang, Graham J. King, Bruce D. L. Fitt, Régine Delourme.

**Data curation:** Yong-Ju Huang.

**Formal analysis:** Yong-Ju Huang, Sophie Paillard.

**Funding acquisition:** Yong-Ju Huang, Bruce D. L. Fitt.

**Investigation:** Yong-Ju Huang.

**Methodology:** Yong-Ju Huang, Vinod Kumar.

**Project administration:** Bruce D. L. Fitt.

**Supervision:** Bruce D. L. Fitt.

**Writing – original draft:** Yong-Ju Huang, Régine Delourme.

**Writing – review & editing:** Yong-Ju Huang, Graham J. King, Bruce D. L. Fitt, Régine Delourme.

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
