## [Decision Letter · Decision Letter 0]

12 Jul 2019

PONE-D-19-15868

Oilseed rape (Brassica napus) resistance to growth of Leptosphaeria maculans in leaves of young plants contributes to quantitative resistance in stems of adult plants

PLOS ONE

Dear Dr Huang,

Thank you for submitting your manuscript to PLOS ONE. After careful consideration, we feel that it has merit but does not fully meet PLOS ONE’s publication criteria as it currently stands. Therefore, we invite you to submit a revised version of the manuscript that addresses the points raised during the review process.

We would appreciate receiving your revised manuscript by 31st July 2019. To enhance the reproducibility of your results, we recommend that if applicable you deposit your laboratory protocols in protocols.io, where a protocol can be assigned its own identifier (DOI) such that it can be cited independently in the future. For instructions see: http://journals.plos.org/plosone/s/submission-guidelines#loc-laboratory-protocols

We look forward to receiving your revised manuscript.

Kind regards,

Harsh Raman, Ph.D

Academic Editor

PLOS ONE

Journal Requirements:

2. Our internal editors have looked over your manuscript and determined that it is within the scope of our Future Crops Call for Papers. This collection of papers is headed by a team of Guest Editors for PLOS ONE. The Collection will encompass a diverse range of research articles on enhanced agronomic production, guaranteeing food security and neglected crop species.  Additional information can be found on our announcement page: https://collections.plos.org/s/future-crops.

If you would like your manuscript to be considered for this collection, please let us know in your cover letter and we will ensure that your paper is treated as if you were responding to this call. If you would prefer to remove your manuscript from collection consideration, please specify this in the cover letter.

Additional Editor Comments:

Overall manuscript is well written. Please format this manuscript properly (currently in different fonts and styles etc) and address comments raised by both reviewers. Here are my further comments

L78-81: Revise sentence

L130: Insert also reference 56 (as it validates). Also, Yudal has QTL for quantitative resistance! Revise accordingly

L172, L274: delete nine lines. It is written in the first line

L188, 323: format headings, use the same font, delete 190 lines

L197-198: You may delete, 'therefore, the condia...........inoculum".

L328: The continuous distributions?

L332: broad sense heritability?

Figure 3: Rearrange the order as A, B and C

Figure 4; Provide high resolution image

Table 3: which version of the reference genome was used. Please describe in methods or in this Table. Physical interval in bp?

Table 4: Please provide physical interval here

Reviewers' comments:

Reviewer's Responses to Questions

**Comments to the Author**

1. Is the manuscript technically sound, and do the data support the conclusions?

Reviewer #1: Yes

Reviewer #2: Yes

2. Has the statistical analysis been performed appropriately and rigorously? 

Reviewer #1: Yes

Reviewer #2: Yes

3. Have the authors made all data underlying the findings in their manuscript fully available?

Reviewer #1: Yes

Reviewer #2: Yes

4. Is the manuscript presented in an intelligible fashion and written in standard English?

Reviewer #1: Yes

Reviewer #2: Yes

5. Review Comments to the Author

Reviewer #1: Manuscript title: Oilseed rape (Brassica napus) resistance to growth of Leptosphaeria maculans in leaves of young plants contributes to quantitative resistance in stems of adult plants (PONE-D-19-15868) by Huang et al.

The authors of this manuscript have attempted to identify the components of quantitative resistance to Leptosphaeria maculans (blackleg) in oilseed rape that are highly desirable in durable resistance breeding for blackleg. However, my main concern is that authors used only one isolate in their investigations, therefore, how the breeders are going to use this information when large number of pathogen races exists and that too keep on changing every year! Whether they should be screening against more prevalent but less virulent races or should they screen against less prevalent but more virulent ones? Overall, the manuscript is suitable for publication in Plos One with some specific points that need to be addressed as outlined below:

Authors chose nine DH lines of a (Darmor-bzh x Yudal) population based upon their crown/stem canker severity assessed in 1995, 1996 and 2007, which was similar in these three years; authors need to provide a reference for this. In their current investigations, authors related the outcomes of glasshouse/controlled environment experiments with the stem canker severity assessed so long ago (1995, 1996 and 1997). My main concern is that why authors did not attempt to retest the stem canker severity under field conditions during their current investigations to validate their claims. There are significant amount of published papers demonstrating that population of L. maculans changes very rapidly, it is likely that the chosen DH lines may behave differently to the existing populations under field conditions to the populations they were exposed to about 11 to 20 years ago. Moreover, there is evidence that quantitative resistance is eroded over time.

In the methods section authors used ascospore inoculum from cultivar, ‘Courage’, need to provide more information regarding whether it was a susceptible or resistant cultivar and/ or the Rlm gene it carries.

It was interesting that authors used visual assessments to determine the growth of the fungus (Gt) down the petiole from the infection site. Generally, such a growth is systemic and often not visible in most instances so how confident authors are that this trait can be used as one of the criterion for determining quantitative resistance. My other concerns is why authors did not include an absolute susceptible check to compare the Gt in DH lines to that of a susceptible line. I understand they have used one susceptible DH line but including a highly susceptible check would have been much valuable.

It is not very clear how many plants per DH line for each replication were used in the first experiment. In the second experiment with 190 DH lines, it appears that two plants per DH lines were used and there were three reps that equates to just six plants for each DH line in total. In my view, the experimental unit is too small to draw useful conclusions. In addition, in the second experiment, authors maintained high humidity for 72 hours, whereas, in the first experiment, this was for 48 hours; authors need to mention the reasons for this inconsistency. This will be helpful for readers who are not familiar with the relative infection ability of ascospores vs pycnidiospores.

In Table 1, authors need to provide the severity rating scale (for example, 0-5 or 0-4 etc.) used in 1995, 1996 and 2007 experiments in France for calculating the stem canker disease index (DI) and bit more details on how the DI was calculated.

On page 13, authors mentioned a wide range in leaf petiole length and total leaf length but they had not included data for the leaf lamina length. It is important for the readers to know the range of leaf lamina length as if the length of the lamina is larger than that of the petiole then the fungus may not be able to reach the petiole, as the maximum distance travelled by the fungus in these investigations is merely 3cm in a susceptible line DY2. In this instance, it is obvious that there will be no correlation between total leaf and petiole length with the distance travelled by the fungus.

I have noticed that authors inoculated the leaves very close to midrib, have they speculated the situation if lesions are far away from the midrib? How will that account for the fungus to reach to the petiole and into the stem?

Fig. 3 need to be reordered from A to C, labels for Y and X-axis for each of the A, B and C need to be provided, though Y-axis label is common for A to C.

Reviewer #2: This is a very well prepared manuscript. I only have a few comments.

Lines 132-134: It would be helpful if the previously-detected QTL within these DH lines and previouls field ratings (supporting their classification from susceptible to resistant) was listed, maybe in the Supplimentary?

Line 208 is missing a full stop between sentences.

Table 3 needs to be cleaned up a bit as the justification makes it hard to align numbers with rows/traits. As the figure is presented in Mb and the support intervals are also given in bp, perhaps the cM positions are expendible?

Lines 471-477: Later in the discussion you draw attention to previous work which tested QTL under controlled conditions using cotyledon infection. While the current study may be the first to do this with true leaf infection, the cotyledon assays still fall in the 'stage-1' of L. maculans growth, and should probably be metioned here rather than later.

6. PLOS authors have the option to publish the peer review history of their article (what does this mean?). If published, this will include your full peer review and any attached files.

Reviewer #1: No

Reviewer #2: No

---

## [Author Response · Author response to Decision Letter 0]

29 Jul 2019

Our response to specific reviewer and editor comments is in the file 'Response to reviewers'.

---

## [Decision Letter · Decision Letter 1]

3 Sep 2019

[EXSCINDED]

Oilseed rape (Brassica napus) resistance to growth of Leptosphaeria maculans in leaves of young plants contributes to quantitative resistance in stems of adult plants

PONE-D-19-15868R1

Dear Dr. Huang,

We are pleased to inform you that your manuscript has been judged scientifically suitable for publication and will be formally accepted for publication once it complies with all outstanding technical requirements.

With kind regards,

Harsh Raman, Ph.D

Academic Editor

PLOS ONE

Additional Editor Comments (optional):

Reviewers' comments:

Reviewer's Responses to Questions

**Comments to the Author**

1. If the authors have adequately addressed your comments raised in a previous round of review and you feel that this manuscript is now acceptable for publication, you may indicate that here to bypass the “Comments to the Author” section, enter your conflict of interest statement in the “Confidential to Editor” section, and submit your "Accept" recommendation.

Reviewer #1: (No Response)

Reviewer #2: All comments have been addressed

2. Is the manuscript technically sound, and do the data support the conclusions?

Reviewer #1: (No Response)

Reviewer #2: Yes

3. Has the statistical analysis been performed appropriately and rigorously? 

Reviewer #1: (No Response)

Reviewer #2: Yes

4. Have the authors made all data underlying the findings in their manuscript fully available?

Reviewer #1: (No Response)

Reviewer #2: (No Response)

5. Is the manuscript presented in an intelligible fashion and written in standard English?

Reviewer #1: (No Response)

Reviewer #2: Yes

6. Review Comments to the Author

Reviewer #1: (No Response)

Reviewer #2: (No Response)

7. PLOS authors have the option to publish the peer review history of their article (what does this mean?). If published, this will include your full peer review and any attached files.

Reviewer #1: No

Reviewer #2: No

---

## [Editor Report · Acceptance letter]

4 Sep 2019

PONE-D-19-15868R1 

Oilseed rape (*Brassica napus*) resistance to growth of *Leptosphaeria maculans* in leaves of young plants contributes to quantitative resistance in stems of adult plants 

Dear Dr. Huang:

I am pleased to inform you that your manuscript has been deemed suitable for publication in PLOS ONE. Congratulations! Your manuscript is now with our production department. 

With kind regards,

on behalf of

Dr. Harsh Raman 

Academic Editor

PLOS ONE